# Mapping of Land Degradation Vulnerability in the Semi-Arid Watershed of Rajasthan, India

**Lal Chand Malav** [1], **Brijesh Yadav** [1,*], **Bhagwati L. Tailor** [1], **Sarthak Pattanayak** [2], **Shruti V. Singh** [3], **Nirmal Kumar** [4], **Gangalakunta P. O. Reddy** [4], **Banshi L. Mina** [1], **Brahma S. Dwivedi** [4] and **Prakash Kumar Jha** [5]

1   ICAR—National Bureau of Soil Survey & Land Use Planning, Regional Centre, Udaipur 313001, India
2   Krishi Vigyan Kendra, Odisha University of Agriculture and Technology, Bolangir 751003, India
3   Krishi Vigyan Kendra, ICAR—Indian Institute of Vegetable Research, Kushinagar 274406, India
4   ICAR—National Bureau of Soil Survey & Land Use Planning, Amravati Road, Nagpur 440033, India
5   Feed the Future Innovation Lab for Collaborative Research on Sustainable Intensification, Kansas State University, Manhattan, KS 66506, USA
*   Correspondence: brijesh.yadav@icar.gov.in

**Abstract:** Global soils are under extreme pressure from various threats due to population expansion, economic development, and climate change. Mapping of land degradation vulnerability (LDV) using geospatial techniques play a significant role and has great importance, especially in semi-arid climates for the management of natural resources in a sustainable manner. The present study was conducted to assess the spatial distribution of land degradation hotspots based on some important parameters such as land use/land cover (LULC), Normalized Difference Vegetation Index (NDVI), terrain characteristics (Topographic Wetness Index and Multi-Resolution Index of Valley Bottom Flatness), climatic parameters (land surface temperature and mean annual rainfall), and pedological attributes (soil texture and soil organic carbon) by using Analytical Hierarchical Process (AHP) and GIS techniques in the semi-arid region of the Bundi district, Rajasthan, India. Land surface temperature (LST) and NDVI products were derived from time-series Moderate-Resolution Imaging Spectroradiometer (MODIS) datasets, rainfall data products from Climate Hazards Group InfraRed Precipitation with Station data (CHIRPS), terrain characteristics from Shuttle Radar Topography Mission (SRTM), LULC from Landsat 9, and pedological variables from legacy soil datasets. Weights derived for thematic layers from the AHP in the studied area were as follows: LULC (0.38) > NDVI (0.23) > ST (0.15) > LST (0.08) > TWI (0.06) > MAR (0.05) > SOC (0.03) > MRVBF (0.02). The consistency ratio (CR) for all studied parameters was <0.10, indicating the high accuracy of the AHP. The results show that about 20.52% and 23.54% of study area was under moderate and high to very high vulnerability of land degradation, respectively. Validation of LDV zones with the help of ultra-high-resolution Google Earth imageries indicates good agreement with the model outputs. The research aids in a better understanding of the influence of land degradation on long-term land management and development at the watershed level.

**Keywords:** analytical hierarchical process; land degradation vulnerability; NDVI; land surface temperature; soil properties

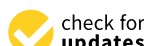



## 1. Introduction

Land is a vital and precious resource to produce food, fiber, fuel, and other ecosystem services for the survival of humans and animals [1,2]. However, the constant pace of degradation and deterioration due to persistent human-induced disturbances and climatic irregularities [3] places livelihood and sustainable progress under acute threat [4]. Land degradation is a major environmental problem all around the world and influences human society and its livelihoods. Globally, the life of around 3.2 billion people totally

depends on degraded lands, and around one-third of the world's lands are affected by land degradation [5,6]. In recent years, land degradation has been considered a pivotal factor in environmental issues and has attracted the attention of all stakeholders [7]. The United Nations General Assembly adopted Sustainable Development Goal 15.3 in September 2015, which focuses on achieving land degradation neutrality (LDN) by implementing the best management practices that reduce the loss of healthy land and maintain or improve the productivity of the land [8,9]. Land degradation can be defined as a spatio-temporal deterioration of physico-chemical and biological properties of land, making it unsuitable for human society, and a deterioration of the soil ecosystem, influencing agricultural production and ecological instability [10,11].

Around 24% of the world's total geographic area (approximately 3500 Mha) is severely affected by land degradation [11,12]. Around 20% of cropland, 10% of grassland, and 30% of forests are under the process of land degradation throughout the world [13]. In India, around 36.7% of total geographical area (TGA) (120.7 Mha) is under different types of land degradation such as soil erosion, soil acidity, soil salinity and alkalinity, and waterlogging [14], and soil salinity and alkalinity alone affect 6.73 Mha in different arid, semi-arid, and sub-humid areas [15]. According to the Indian Space Research Organization (ISRO), land degradation accounts for around 29.32% of the TGA of India. It covers 96.4 Mha of agricultural, forest, and non-forest land spread across the country [16]. India joined the Bonn Challenge and the United Nations Decade on Ecosystem Restoration 2021–2030 to maximize ecological and economic advantages from the restoration of degraded ecosystems, pledging to rehabilitate 26 Mha of degraded land by 2030 [17].

The problem of land degradation is especially severe in arid and semi-arid areas of the country, such as the state of Rajasthan. Land degradation affects 67% of Rajasthan's land, where wind erosion contributes to the maximum percentage (44.2%), and water erosion (11.2%), vegetal degradation (6.25%), and salinization (1.07%) are the next most common forms of degradation. Chambal ravines in the state of Rajasthan are perhaps among the worst physically degraded lands, as cultivated fertile lands were engulfed by ravines and rendered unsuitable for agricultural activities [18]. The Chambal ravines are very typical as they are deep to very deep (>20 m) and are devoid of any kind of vegetation, with ravines and gullies being the typical forms of degradation [19]. For the development of effective strategies to minimize and lessen the effects of land degradation, it is a prerequisite to understand the process of land degradation, including the causes and its consequences for major functions of the ecosystem and the proper identification of the affected area and the regions at high risk.

Modeling and assessing the vulnerability of land degradation play a pivotal role in land degradation neutrality planning and prioritization processes and in fulfilling targets for restoration. Assessment of land degradation requires various information such as climate, soil properties, topography, land use, etc. Several techniques are being adopted in monitoring and evaluating the area, rate, and type of land degradation. A survey using satellite images overcomes the time-consuming and expensive traditional survey, particularly in areas tough to assess [20]. Geospatial techniques such as remote sensing (RS) and geographic information system (GIS) play an important role in the assessment and monitoring of land degradation vulnerability. Satellite imageries with precise spatial and spectral resolution are excellent resources for detecting, mapping, and monitoring various degradation kinds and issues in a rapid, consistent, reliable, and cost-effective manner [21–25].

The integrated use of geospatial techniques with the multi-criterion decision analysis (MCDA) method is the most feasible option to assess and map land degradation vulnerability. This MCDA technique has numerous applications in multiple areas such as groundwater potential mapping, crop suitability zonation, and land degradation vulnerable mapping. It is mostly used to solve complex problems by breaking them up into sections, then solving and integrating each section to obtain the ultimate results. The AHP, which was first developed by Saaty (1980), is the most widely used multi-criterion

decision method for the mapping of vulnerable zones [26,27]. Decisions may be made using this strategy based on judgements, hierarchical structure, and accurate perception, all of which have a dominant influence on the final decision [27,28]. The AHP approach is a widely recognized, basic, and well-structured decision-making technique. Few research findings have been generated by other researchers [12,13,29] with respect to the assessment and mapping of land degradation vulnerability zones (LDVZ) based on AHP and GIS modeling approaches and their validation with Google Earth imageries. Considering the importance of land degradation vulnerability assessment through remote sensing and GIS and AHP approaches, the present study was carried out in the semi-arid region of Rajasthan, western India. In the present study area, water erosion is the most important cause of land degradation due to favorable erosion geology, vegetal degradation, and the perennial Chambal River. Despite this fact, so far, no studies have been carried out in this area to assess and prepare a land degradation susceptibility map. The core objectives of the study are to (i) characterize the terrain, climatic, vegetative, and pedological variables of the watershed and (ii) identify the most vulnerable areas to land degradation using remote sensing and geospatial techniques. Furthermore, the research provides important information for long-term land use management and development.

## 2. Materials and Methods

### 2.1. Study Area

The Chanda Kalan Watershed is in the Bundi district of Rajasthan, western India, and it lies between latitude 25°41′ N to 25°46′ N and longitude 76°16′ E to 76°22′ E. Geographically, it covers an area of 2629 hectares (Figure 1). The watershed falls within the Northern Plain (and Central Highlands) including Aravalli, a hot semi-arid eco-region (4.2) denoted as an agro-ecological sub-region (AESR). The climate of the study area is semi-arid with an average annual rainfall of 681 mm, in which the southwest (SW) monsoon contributes roughly 90% of the rainfall. The altitude ranges from 187 to 459 m from the mean sea level (MSL). The watershed is mainly drained by the Chambal River and its tributaries. Major soils are deep brown loamy and brown clayey. The important crops cultivated in the study area are wheat, maize, rapeseed, soybean, paddy, etc. Geologically, the watershed is exposed by rock formations belonging to the Vindhyan Super Group. Vindhyan sedimentary sequences have occupied a major part of the watershed. The Bhander Group of the Vindhyan Super Group and their formations (Upper Bhander shale, Balwan Limestone, Maihar Sandstone) are well exposed in the study area [30]. The watershed has a systematic drainage system, and most of the study area is drained by the southwest to northeast flowing Chambal River and its tributaries. The aquifer area formed in the watershed comes under younger alluvium.

### 2.2. Dataset Used

In the current study, eight thematic layers were considered to identify the land degradation vulnerable zones, including Moderate Resolution Imaging Spectroradiometer (MODIS) Normalized Difference Vegetation Index (NDVI), MODIS land surface temperature (LST), Climate Hazards Group Infrared Precipitation with Station data (CHIRPS) rainfall, land use/land cover (LULC), Topographical Wetness Index (TWI), Multi-Resolution Index of Valley Bottom Flatness (MRVBF), and soil texture and soil organic carbon. The Landsat 9 images and Shuttle Radar Topography Mission (SRTM) DEM data were collected from the United States Geological Survey (USGS) website (https://earthexplorer.usgs.gov, accessed on 10 February 2022). The CHIRPS rainfall, MODIS NDVI, and MODIS LST products were downloaded for the period of 10 years (2011–2020) using Google Earth Engine. Soil organic carbon data were downloaded from Soil Grids (https://soilgrids.org/, accessed on 11 February 2022). In addition, soil texture data were collected from the ICAR—National Bureau of Soil Survey and Land Use Planning, Nagpur, at 1:250,000 scale. Various datasets and their specifications are summarized in Table 1.

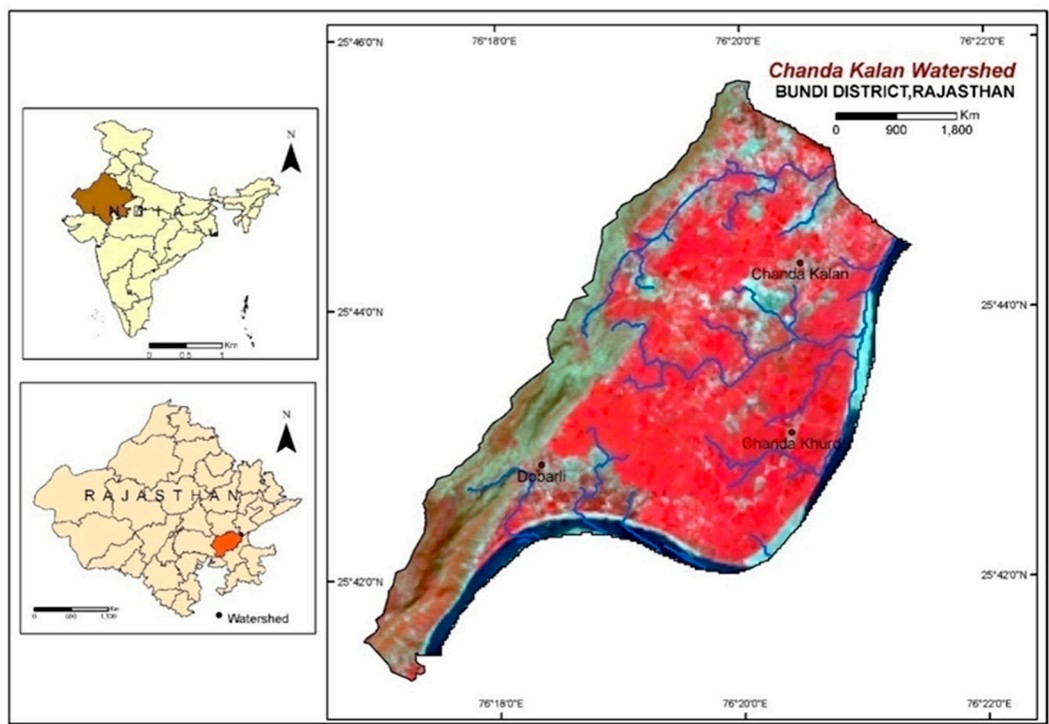

**Figure 1.** Location map of the study area.

**Table 1.** Datasets and their specifications.

| S. No. | Dataset | Variable | Temporal Resolution | Spatial Resolution | Temporal Coverage |
|---|---|---|---|---|---|
| 1 | MODIS MOD13Q1 | NDVI | 16 days | 250 m | 2011–2020 |
| 2 | MODIS MOD11A2 | LST | 8 days | 1 km | 2011–2020 |
| 3 | SRTM DEM | Elevation | - | 30 m | - |
| 4 | Soil Grids 250 m | Soil organic carbon | - | 250 m | - |
| 5 | CHIRPS | Rainfall | - | 5 km | 2011–2020 |
| 6 | SRM data, NBSS&LUP | Soil texture | - | 2.5 km | - |

### 2.3. Processing of Data

#### 2.3.1. Processing of Terrain Parameters

The SRTM DEM was downloaded and reprojected to Universal Transverse Mercator (UTM), 43 N coordinate system, and filled in QGIS. After that, the filled DEM was used to produce TWI and MRVBF of the watershed. The TWI is widely used to evaluate the impact of topography on different hydrological processes, and it is considered an important indicator of the wetness conditions of a particular region [31]. TWI depicts the water accumulation tendency of a region [32]. Therefore, with respect to land degradation, a higher value of the TWI indicates less vulnerability to degradation or water erosion, and vice versa. In this study, TWI was computed in the SAGA GIS using the following equation:

$$TWI = \ln(As/\tan \beta ) \tag{1}$$

where As is the area of the ascending slope and β is the gradient of the slope.

The flatness and lowness of valley bottoms are measured by an index called the Multi-Resolution Index of Valley Bottom Flatness (MRVBF). A higher value of MRVBF denotes a flatter valley with higher deposition, and vice versa. MRVBF values range from 0 to a positive integer value. In this study, MRVBF was computed in the SAGA GIS.

### 2.3.2. Processing of Climate Parameters

The combined effects of climate, physical processes, and land use practices are often the cause of land degradation. Rainfall is the most significant factor in land degradation, and it has a direct impact on the detachment of soil particles and migration of eroded sediment [33]. As a result, it is recognized as a major factor in assessing land degradation. In this study, CHRIPS-based rainfall products of 5 km spatial resolution were downloaded for 10 years (2011–2020) using Google Earth Engine and reprojected from the Geographic Coordinate System (GCS) to the UTM 43N coordinate system in QGIS. The downloaded products were resampled to 30 m resolution in QGIS by using the bilinear interpolation technique. The intensity and distribution of land surface temperature are directly linked to the vegetative condition of a region [13]. Therefore, land surface temperature is considered as an important indicator of land degradation. In the present study, MODIS MOD11A2 products for the period of 10 years (2011–2020) were downloaded using Google Earth Engine. The data were converted to degrees Celsius (°C) by using Equation (2).

$$LST = 0.02 \times DN - 273.15 \tag{2}$$

Subsequently, the datasets were reprojected from the sinusoidal coordinates system to the geographical coordinate system and resampled to 30 m by using the bilinear interpolation technique in QGIS. Finally, the thematic layer was classified into five subclasses: <32.70 °C, 32.70–33.30 °C, 33.30–33.91 °C, 33.91–34.52 °C, and >34.52 °C.

### 2.3.3. Processing of Vegetation Parameters

Vegetal degradation is a direct indicator of land degradation. Therefore, LULC and NDVI were taken as important thematic layers for assessing land degradation. The LULC map was prepared from the downloaded Landsat 9 images using supervised classification in the QGIS environment. In the present study, MODISMOD13Q1 NDVI products were downloaded for a period of 10 years (2011–2020) using Google Earth Engine. NDVI, which is a dimensionless index, depicts the difference of reflectance between near-infrared and red bands and can be used to analyze vegetative greenness over an area. It ranges from −1 to +1, where low NDVI values indicate stressed vegetation and higher values indicate healthy vegetation. Temporal smoothing of the NDVI time series data was carried out with the Savitzky–Golay (SG) filter [34]. SG filter fits a polynomial function based on a weighted least squares regression approach. The processing was executed in Google Earth Engine and downloaded. It was then reprojected from GCS to the UTM 43N coordinate system in QGIS. Datasets were resampled to 30 m resolution by using the bilinear interpolation technique in the QGIS. The layer was classified into six subclasses: <0.15, 0.15–0.20, 0.20–0.25, 0.25–0.30, 0.35–0.40, and >0.40.

### 2.3.4. Processing of Soil Parameters

Soil organic carbon data downloaded from Soil Grids were reprojected to the geographical coordinate system and resampled to 30 m resolution in QGIS. The layer was classified into five subclasses: <122.5, 122.5–176.5, 176.5–230.5, 230.5–284.5 and >284.5 decigram/kg. Soil texture data were taken from NBSS&LUP and resampled to 30 m in the QGIS environment. Soil texture data were classified into three classes, namely, fine loamy, clayey, and rock outcrops. The detailed methodology is given in Figure 2.

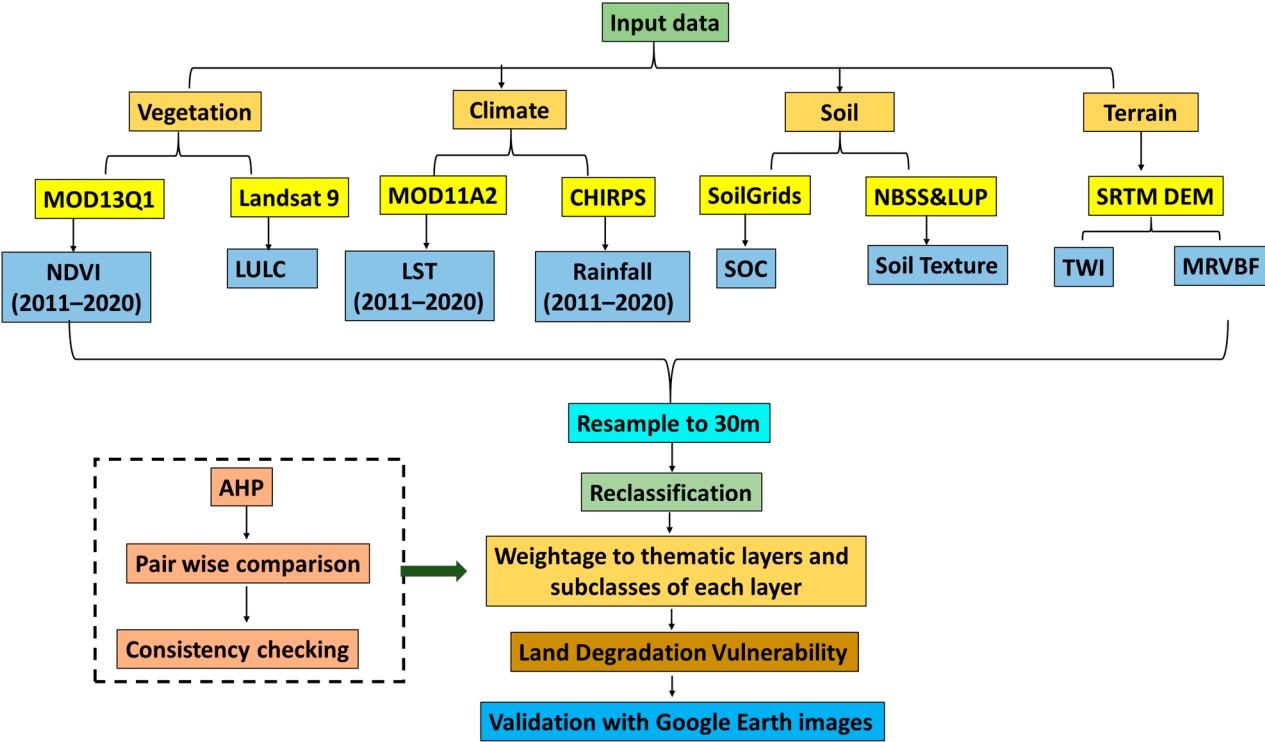

**Figure 2.** Methodology followed in the present study.

### 2.4. Analytical Hierarchical Process (AHP)

The most widely used and well-known GIS-based method for demarcating land degradation vulnerability zones is MCDA using the AHP technique. To make an organized decision of priorities, we need to make comparisons and a scale of numbers that show how much more important one parameter is in comparison to another in terms of the criterion being compared. The AHP is a pairwise comparison assessment theory, where parameters are compared with each other using Saaty's scale of relative importance (Table 2) [31,35].

**Table 2.** Saaty's 1–9 scale of relative importance in AHP.

| Scale | Importance |
|:---:|:---:|
| 1 | Equal significance |
| 2 | Intermediate between 1 and 3 |
| 3 | Moderate significance |
| 4 | Intermediate between 3 and 5 |
| 5 | Strong |
| 6 | Intermediate between 5 and 7 |
| 7 | Very strong |
| 8 | Intermediate between 7 and 9 |
| 9 | Maximum importance |

The relative weight of each variable was determined by a knowledge-based spatial decision support system and referring to the literature [13,29]. We selected LULC as the first significant layer since LULC changes are one of the main human-induced activities affecting the land degradation of a region. NDVI was selected as the second most important parameter in the hierarchy as it is the most significant indicator of vegetal degradation. The soil texture was chosen as the third element in the hierarchy because soil erosion is directly controlled by the size and distribution of soil particles. The LST was selected as the fourth layer in the hierarchy, and it was mainly based on the assumption that higher LST zones have low vegetation cover compared to low LST. Other remaining layers were assigned

lower order in the hierarchy. Consequently, all layers were compared to one another in a pair-wise comparison matrix.

### 2.5. Consistency Analysis

To authenticate the decision on the pair-wise comparison of the thematic layers and their sub-classes, the consistency ratio (CR) was utilized [28]. For computing the CR, the following equation was used:

$$CR = \frac{CI}{RCI} \tag{3}$$

where RCI stands for Random Consistency Index, and its values are based on Saaty's standard (Table 3). CI indicates consistency index, which was computed using the following equation:

$$CI = \frac{(\lambda max - n)}{(n - 1)} \tag{4}$$

where $\lambda_{max}$ is the principal eigenvalue and n is the total number of thematic layers used in the study.

**Table 3.** Saaty's Random Consistency Index.

| N | 1 | 2 | 3 | 4 | 5 | 6 | 7 | 8 | 9 | 10 | 11 |
|---|---|---|---|---|---|---|---|---|---|----|----|
| RCI | 0 | 0 | 0.58 | 0.9 | 1.12 | 1.24 | 1.32 | 1.41 | 1.45 | 1.49 | 1.51 |

N, Order of the matrix; RCI, Random Consistency Index.

A CR value ≤0.10 is acceptable to conduct a weighted overlay analysis using AHP. If the CR is >0.10, the judgement must be revised to identify the cause of the inconsistency and fix it until the CR ≤0.10 is reached.

### 2.6. Mapping of Land Degradation Vulnerability Zones

In GIS-based modeling, AHP-based weights were given to thematic layers and their sub-classes to demarcate the land degradation vulnerability (LDV) zones. In the present study, the following equation was used to delineate the land degradation vulnerability map:

$$\begin{aligned} LDV = \ &LULCCwi \times LULCSCwi + NDVICwi \times NDVISCwi + STCwi \times STSCwi + \\ &LSTCwi \times LSTSCwi + TWICwi \times TWISCwi + MARCwi \times MARSCwi + \\ &SOCCwi \times SOCSCwi0.05 \times MRVBFCwi + MRVBFSCwi \end{aligned} \tag{5}$$

where LULC, NDVI, ST, LST, TWI, MAR, SOC, and MRVBF indicate land use/land cover, Normalized Difference Vegetation Index, soil texture, land surface temperature, Topographic Wetness Index, mean annual rainfall, soil organic carbon, and Multi-Resolution Index of Valley Bottom Flatness, respectively; Cwi is the class weight and SCwi is the sub-class weight. The generated LDV map was classified into five classes, namely, very low, low, moderate, high, and very high. Using the ultra-high resolution Google Earth imagery of 2022, the very high and high LDV classes were validated at five randomly selected sites. Finally, validation of the results was performed using the ROC curve generated from the site selected from the Google Earth image. The area under the curve (AUC) was estimated from the ROC curve and its values range from 0.5 to 1. The AUC value closer to 1 implies great model performance, whereas a value near to 0.5 indicates poor prediction accuracy.

## 3. Result

### 3.1. Input Parameters and Their Variability

TWI of the watershed ranged between 1.46 and 25.94, as illustrated in Figure 3a. Five classes—less than 6.13, 6.13–9.45, 9.45–12.77, 12.77–16.10, and more than 16.10—were generated after reclassification. Nearly 64% of the district area came under the first and second subclass of TWI and the remaining 36% came under other subclasses. TWI values show the parts in the study area that are more prone to water erosion.

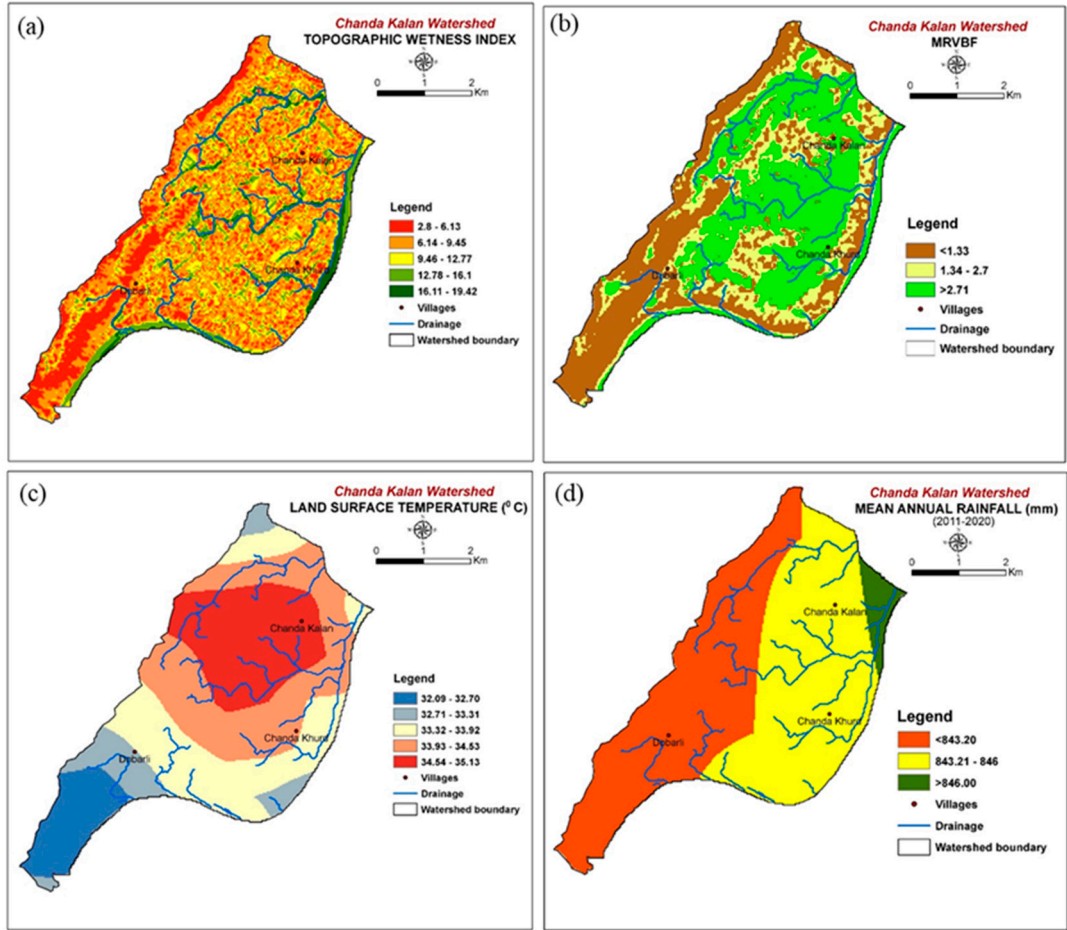

**Figure 3.** Thematic layers: (**a**) TWI, (**b**) MRVBF, (**c**) LST, (**d**) MAR.

In this study, MRVBF is classified into three classes, namely, first class (<1.33), second class (1.33–2.77), and third class (>2.77) (Figure 3b). The highest percentage of the study area comes under third class (43%), followed by first class (35%) and second class (22%). The land surface temperature of the study area divides the whole area into six subclasses: <32.7 °C, 32.70–33.30 °C, 33.30–33.91 °C, 33.91–34.52 °C, and >34.52 °C (Figure 3c). The highest area 801.54 ha (30.51%) comes under subclass LST 33.91–34.52 °C, followed by subclass LST >34.54 °C occupying 630.9 ha (24.02%), and the lowest area comes under subclass LST <32.7 °C of an area 271.89 (10.35%). Nearly 54.53% of the study area came under LST values >34.52 °C and 33.91–34.52 °C, which lies in the central part of the watershed, and the remaining 45.37% of the area came under other subclasses. The analysis of decadal (2011–2020) mean annual rainfall trends shows that the study area is divided into two subclasses (Figure 3d), where northern, northeastern, eastern, and the majority of the central area received relatively higher annual rainfall (>843.10 mm), covering an area of 1760.04 ha that accounts for 66.88% of the total area, while southwestern and southern parts received relatively less annual rainfall (<843.1 mm), covering an area of 871.65 ha that accounts for 33.12% of the total study area.

The spatial analysis of the mean NDVI values from 2011 to 2022 divides the whole area of the Chanda Kalan Watershed (2626.56 ha) into six subclasses: 0.15–0.20, 0.20–0.25, 0.25–0.30, 0.30–0.35, 0.35–0.40, and >0.40 (Figure 4a). The highest area, 926.19 ha (35.26%), comes under the low vegetal degradation category with NDVI values of 0.30 to 0.35, followed by an area of 851.13 ha (32.4%) covering maximum greenness with very low vegetal degradation with NDVI values of 0.35–0.40. The lowest area, 40.5 ha (1.54%), falls under the very severe vegetal degradation category with NDVI values of 0.15 to 0.2. Nearly 67.66% of the study area came under NDVI values 0.30–0.35 and 0.35–0.40, which lies in the southeast and southwest watershed, and the remaining 32.34% of the area came under other subclasses. The LULC map was prepared from Landsat 9 using supervised classification classes (Figure 4b). The study area of Chanda Kalan Watershed is divided into seven classes, namely, Agriculture, Bare Ground, Shrubs/Scrub, Open Forest, Ravines, Built Up, and Water Bodies, based on the LULC map, where the area under Agriculture covers the highest area of the

watershed, which lies in the central and northeast part of the watershed. The soil textural map of the study area divides the whole area into three textural classes, namely, (i) clayey, (ii) fine loamy soil, and (iii) rock outcrops (Figure 4c). Most of the study area comes under clayey soil, particularly the central and the southeastern parts, while the northern and northeastern areas come under the fine loamy soil category and the southwestern area comes under the rock outcrop category. The study area is divided into five soil organic carbon class for soil depth of 0–15 cm based on soil organic carbon content (decigram/kg): <122.5, 122.5–176.5, 176.5–230.5, 230.5–284.5, and >284.5 (Figure 4d). The maximum area of 1707.84 ha covering 65.26% comes under the subclass with SOC content <122.5 decigram/kg and lies in the northwest and central part of the watershed, while the minimum area with <284.5 decigram/kg covers 57.24 ha (2.19%) and lies in the lower part of the watershed.

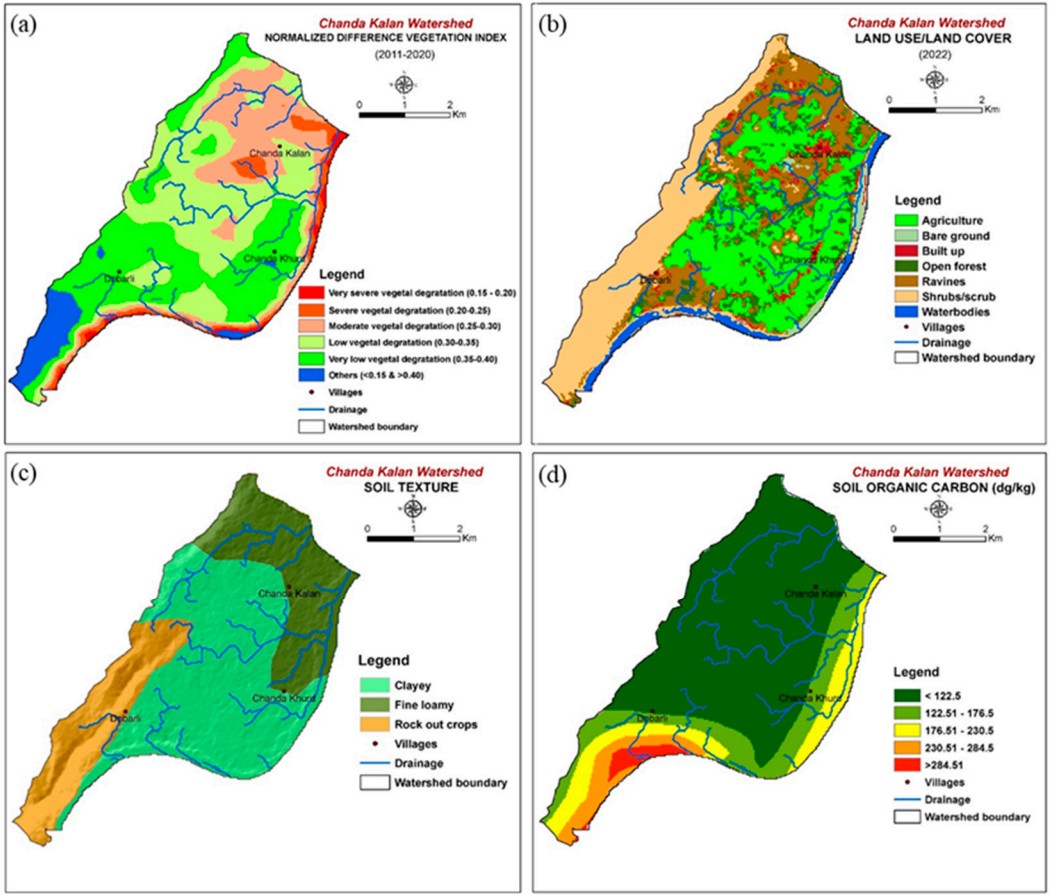

**Figure 4.** Thematic layers: (**a**) NDVI, (**b**) LULC, (**c**) soil texture, (**d**) soil organic carbon.

### 3.2. Land Degradation Vulnerability

Consistency ratios for each thematic layer (Table 4), normalized matrix (Table 5), and subcategories of each thematic layer (Table 6) were calculated before the integration of thematic layers. The results revealed that the judgement matrices utilized in the investigation were accurate (CR < 0.10) and had reasonable consistency. The reclassified thematic layers are combined using the weighted overlay approach based on their respective weight. In this study, five LDVZ categories, namely, very low, low, moderate, high, and very high, were identified through the AHP- and GIS-based modeling approach. The quantile breaks were used for the above classification of the integrated product. The results represent that about 1444.68 hectares of the total study area (55%) are under very low to low classes of land degradation vulnerability, and these lands covered almost half the area of the watershed (Figure 5). About 530.01 hectares (20.52%) of the watershed came under the moderate class of the LDVZ and covered mainly southern to southeastern parts of the watershed. High and very high classes of LDVZ zones covered about 607.95 hectares (23.54%) of the watershed. These classes covered the area mostly the northern and somewhat central and southern parts of the study area. These two classes showed high to very high severity of land degradation, such as ravines and gullies.

**Table 4.** AHP pairwise comparison matrix for thematic layers.

|  | LULC | NDVI | ST | LST | TWI | MAR | SOC | MRVBF | Weight | CR |
|---|---|---|---|---|---|---|---|---|---|---|
| LULC | 1 | 3 | 5 | 5 | 7 | 7 | 8 | 9 | 0.38 | 0.098 |
| NDVI | 0.3 | 1 | 2 | 5 | 5 | 6 | 7 | 8 | 0.23 | |
| ST | 0.2 | 0.5 | 1 | 3 | 3 | 4 | 6 | 7 | 0.15 | |
| LST | 0.2 | 0.2 | 0.3 | 1 | 2 | 3 | 3 | 4 | 0.08 | |
| TWI | 0.1 | 0.2 | 0.3 | 0.5 | 1 | 2 | 2 | 3 | 0.06 | |
| MAR | 0.1 | 0.2 | 0.3 | 0.3 | 0.5 | 1 | 2 | 3 | 0.05 | |
| SOC | 0.1 | 0.1 | 0.2 | 0.3 | 0.5 | 0.5 | 1 | 2 | 0.03 | |
| MRVBF | 0.1 | 0.1 | 0.1 | 0.3 | 0.3 | 0.3 | 0.5 | 1 | 0.02 | |

LULC, land use/land cover; NDVI, Normalized Difference Vegetation Index; ST, soil texture; LST, land surface temperature; TWI, Topographic Wetness Index; MAR, mean annual rainfall; SOC, soil organic carbon; MRVBF, Multi-Resolution Index of Valley Bottom Flatness.

**Table 5.** Normalized matrix for thematic layers.

|  | LULC | NDVI | Texture | LST | TWI | MAR | SOC | MRVBF |
|---|---|---|---|---|---|---|---|---|
| LULC | 0.44 | 0.56 | 0.54 | 0.32 | 0.36 | 0.29 | 0.27 | 0.24 |
| NDVI | 0.15 | 0.19 | 0.22 | 0.32 | 0.26 | 0.25 | 0.24 | 0.22 |
| Texture | 0.09 | 0.09 | 0.11 | 0.19 | 0.16 | 0.17 | 0.20 | 0.19 |
| LST | 0.09 | 0.04 | 0.04 | 0.06 | 0.10 | 0.13 | 0.10 | 0.11 |
| TWI | 0.06 | 0.04 | 0.04 | 0.03 | 0.05 | 0.08 | 0.07 | 0.08 |
| MAR | 0.06 | 0.03 | 0.03 | 0.02 | 0.03 | 0.04 | 0.07 | 0.08 |
| SOC | 0.06 | 0.03 | 0.02 | 0.02 | 0.03 | 0.02 | 0.03 | 0.05 |
| MRVBF | 0.05 | 0.02 | 0.02 | 0.02 | 0.02 | 0.01 | 0.02 | 0.03 |

**Table 6.** Weighting of sub-classes.

| Thematic Layer | Subclass | Weight | CR |
|---|---|---|---|
| LULC | Ravines | 0.511 | 0.085 |
| | Bare ground | 0.292 | |
| | Shrub/scrub | 0.097 | |
| | open forest | 0.062 | |
| | Agriculture | 0.039 | |
| NDVI | 0.15–0.20 | 0.449 | 0.095 |
| | 0.20–0.25 | 0.275 | |
| | 0.25–0.30 | 0.120 | |
| | 0.30–0.35 | 0.076 | |
| | 0.35–0.40 | 0.050 | |
| Soil texture | >0.40 | 0.032 | 0.062 |
| | Rock outcrops | 0.633 | |
| | Fine loamy | 0.260 | |
| | Clayey | 0.106 | |
| LST | <32.7 | 0.062 | 0.026 |
| | 32.70–33.30 | 0.099 | |
| | 33.30–33.91 | 0.161 | |
| | 33.91–34.52 | 0.262 | |
| | >34.52 | 0.416 | |
| TWI | <6.13 | 0.520 | 0.08 |
| | 6.13–9.45 | 0.220 | |
| | 9.45–12.77 | 0.149 | |
| | 12.77–16.10 | 0.073 | |
| | >16.10 | 0.038 | |
| MAR | <843.2 | 0.082 | 0.042 |
| | 843.2–846 | 0.343 | |
| | >846 | 0.575 | |
| SOC | <122.5 | 0.487 | 0.045 |
| | 122.5–176.5 | 0.256 | |
| | 176.5–230.5 | 0.133 | |
| | 230.5–284.5 | 0.081 | |
| | >284.5 | 0.044 | |
| MRVBF | <1.33 | 0.633 | 0.062 |
| | 1.33–2.70 | 0.260 | |
| | >2.70 | 0.106 | |

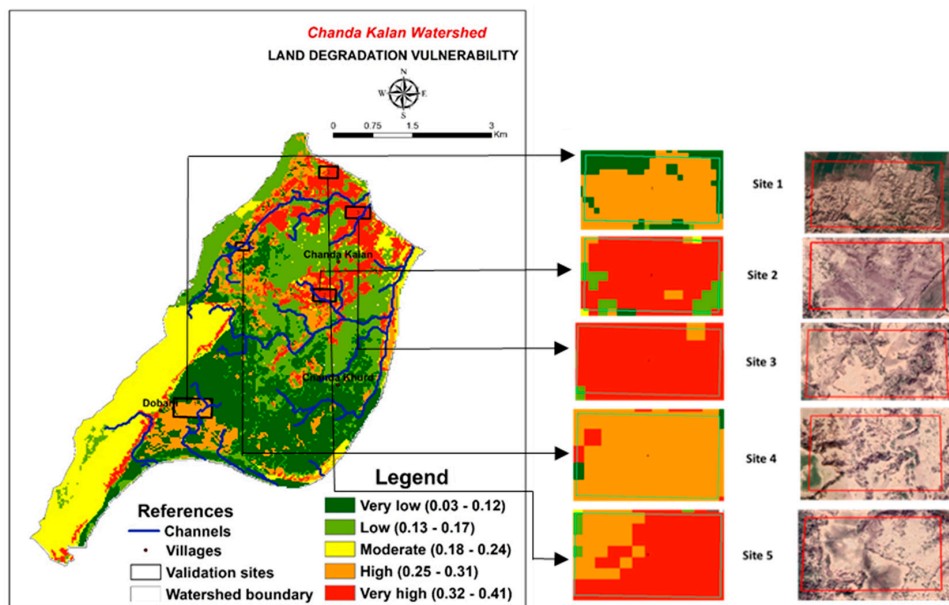

**Figure 5.** Land degradation vulnerable zones and validation with Google Earth images (Dark green color indicates low vulnerability and deep red indicates higher vulnerability).

### 3.3. Validation of Land Degradation Vulnerability Zones

Validation of LDVZ of the study area was conducted by the visual validation method. In this process, validation was performed with the help of high-resolution Google Earth images. Five sites from the degraded part of the study area were validated with the high-resolution Google Earth images. The visual assessment of high and very high classes using high-resolution Google Earth images of growing season of 2022 indicated that the degree of land degradation (ravines, gullies, and bare grounds) in the selected watershed is in accordance with the outcomes of the model used in this study (Figures 5 and 6).

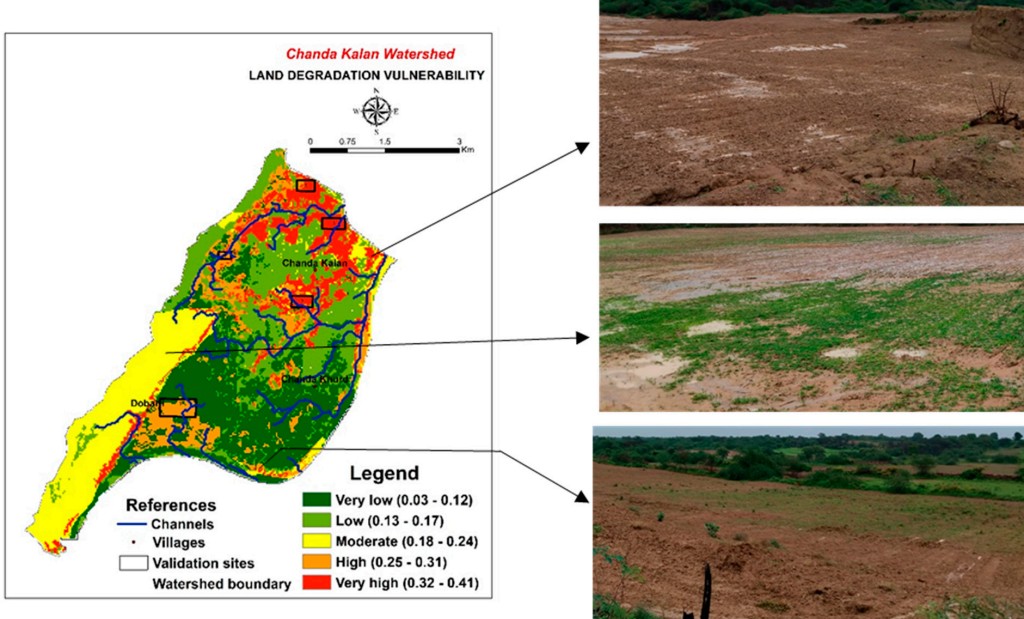

**Figure 6.** Validation of land degradation vulnerable zones with field photographs.

Figure 7 shows the ROC curve of the LDVZ map generated using the AHP method. The AUC value of the ROC curve was found to be 86%. Hence, it was determined that the AHP model derives reasonable results in predicting land degradation vulnerability zones in the study area.

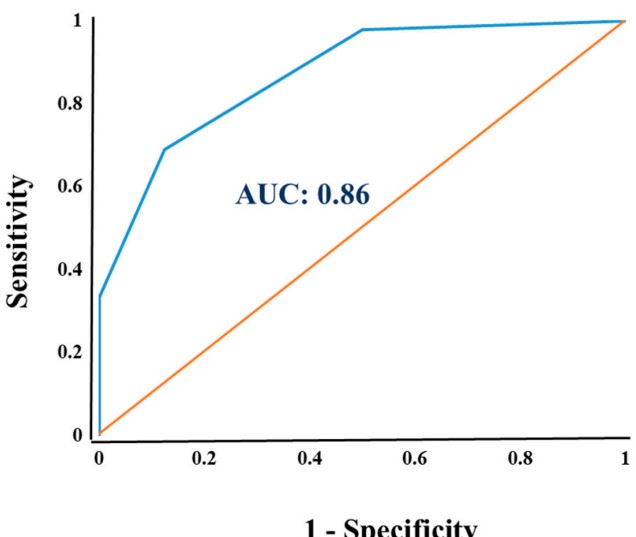

**Figure 7.** ROC curve of the LDVZ map using AHP model.

## 4. Discussion

Land degradation is generally considered one of world's most serious environmental issues. In India, the western state of Rajasthan is a part of the Thar Desert, where degradation is a severe issue. The Chambal River valley in the state of Rajasthan is one of the severely affected regions in the country, where gully erosion/ravines have major physical and economic implications [18,19,36]. For sustainable agricultural planning and development, the identification of vulnerable hotspots to soil/water erosion is the need of the hour. Therefore, the present research was carried out to identify hot spots of land degradation in a small watershed using an AHP- and GIS-based modeling approach. Previous research has found that only a few variables play an important role in the assessment of land degradation [13,37]. In the present investigation, LULC, NDVI, TWI, MRVBF, LST, MAR, soil texture, and SOC were considered for the mapping of land degradation vulnerable zones. LULC and NDVI were taken as the most influential layers for land degradation vulnerability. Land use/land cover implies man-made and natural modification of the land surface, and it is a major cause of land degradation [38,39]. The NDVI has long been recognized as a useful measure for determining the greenness of flora and it is well accepted in science that a decrease in NDVI is a sign of land degradation and is closely linked to climatic conditions [40,41]. The most basic soil physical property, on the other hand, is soil texture, which impacts hydraulic properties and surface soil loss [42].

LST is an important parameter in the semi-arid region as it is directly linked with soil moisture availability and indirectly linked with the flora conditions of the study area [43,44]. A rise in the LST might result in a reduction in vegetative greenness and an increase in land degradation. Increased rainfall during the monsoon season increases the risk of topsoil loss by higher water velocity, which causes more soil erosion [37]. SOC is a universal biomarker of soil degradation since its decrease may have severe consequences for soil-derived ecosystem services [45]. Similarly, TWI is one of the most important terrain parameters and plays a significant role in assessing land degradation vulnerability. A higher TWI value is associated with good vegetation cover, and vice versa [46]. As a result, vegetation cover promotes infiltration, reduces surface runoff, and thus greatly delays the incidence of soil erosion [47]. Therefore, thematic layers were given weights based on their importance. The AHP model assigned the weightage of each factor, i.e., LULC (0.38), NDVI (0.23), soil texture (0.15), LST (0.08), TWI (0.06), MAR (0.05), SOC (0.03), and MRVBF (0.02). The higher the index value, the more exposed the area is to land degradation, whereas the lower the value, the less vulnerable it is. Consistency ratios for each thematic layer and subclasses of each thematic layer were calculated before the integration of thematic layers. The computed CR value was less than 0.1, which shows that all the parameters' assumptions about their impact on soil erosion are valid.

Research findings showed that five land degradation vulnerability zones (LDVZ) namely, very low, low, moderate, high, and very high, were identified in the study area. Very low, low, moderate, high, and very high classes covered 27%, 29%, 20%, 11%, and 12% of the area of the watershed, respectively. Parmar et al. (2021) [48] also conducted a study to assess land degradation vulnerability using the geospatial technique in the Kutch district of Gujarat, India, and the results revealed that 67% of the land area has high vulnerability to land degradation, and 27% of the area falls under

the moderate class. Similarly, an assessment of potential land degradation using the geospatial technique and multi-influencing factor technique was carried by Senapati et al. (2020) [49] in the Akarsa Watershed, West Bengal, and they also classified the study area into five land degraded zones. The analysis revealed that the very low to low categories of land degradation vulnerability covered almost half of the area of the watershed, and this portion of the study area is associated with good vegetative coverage with open forest, very low vegetative degradation, adequate rainfall (843.21–846 mm), and well-drained soils deep in nature with clayey texture.

All the environmental covariables are linked with each other, e.g., adequate rainfall positively correlates with NDVI and LULC and a good amount of LULC links with optimum SOC content and better soil health, which directly relate to a lower chance of land degradation [50]. The moderate class of LDVZ was related to very less vegetative coverage in the scrub/shrub class of LU/LC, with normal rainfall (<843.20 mm) and rock outcrops. This class also represented a low to medium MRVBF value with low TWI. High and very high classes of LDVZ covered about 607.95 hectares (23.54%) of the watershed. These two classes showed high to very high severity of land degradation, such as ravines and gullies. This section of the watershed had no or very little vegetation cover, higher rainfall (>846 mm), high valley bottom flatness, higher LST, and clay to fine loamy texture soils with low soil organic carbon. In this research, the AHP- and GIS-based modeling approach showed its potential for the assessment of vulnerability to land degradation by compiling different parameters. Validation of LDVZ was carried out with the help of Google Earth images of high resolution and the results were very well in agreement with the AHP–GIS model-based approach. Similar work has been conducted by several researchers [12,13,29,51–54] with respect to assessment and mapping of LDVZ based on an AHP and GIS modeling approach and their validation with Google Earth imageries.

This study has identified areas that are more prone to land degradation, which can help prioritize and implement soil water conservation practices to reduce the consequences of degradation. Farmers should be encouraged to grow cover crops and crop rotation practices to maintain soil quality over time. Farmers should maintain crop residue and biomass over soil surface after harvesting to avoid exposing the topsoil. Furthermore, the findings of this research may be useful in developing better soil and water management policies. Although this study was carried out at the watershed level, it should be replicated to the sub-district or district level. The pedological parameters used in this study are available at coarse resolution, which caused some challenge and gaps in the results. Future research should concentrate on high-resolution satellite and soil survey data to delineate degradation zones with higher accuracy.

## 5. Conclusions

In the study, LULC, NDVI, soil texture, LST, MAR, TWI, SOC, and MRVBF were considered major contributing factors in the identification of land degradation vulnerability zones through the GIS- and AHP-based model. The AHP- and GIS-based modeling shows that about 607.95 hectares of the total study area are in the high and very high categories of LDV, and 530.01 hectares are in the moderate LDV category. Validation of moderate, high, and very high LDV classes using high-resolution Google Earth imagery demonstrates that the degree of land degradation features of Google Earth imagery of the selected study area was in agreement with the AHP–GIS model-based approach. This study demonstrates the potential of high-resolution satellite data and the robustness of GIS-based spatial modeling in obtaining accurate, reliable, and cost-effective results for the assessment of land degradation in semi-arid ecosystems. The prevalence and severity of LDV were determined using AHP- and GIS-based modeling, which will be extremely useful in recommending soil conservation and management measures that are suited for each site, particularly in highly and extremely vulnerable regions, for long-term land resources management. These data were derived from satellite data that could cause some challenges and gaps in the results. Therefore, macro- and micro-scale observations are required to account for the high environmental variability and to distinguish between the influences of anthropogenic actions and climate variability on land degradation processes.

**Author Contributions:** Conceptualization, B.Y. and L.C.M.; methodology, G.P.O.R. and N.K.; software, B.L.T. and L.C.M.; validation, S.P. and B.L.M.; formal analysis, B.Y. and L.C.M.; investigation, B.Y.; resources, B.S.D.; data curation, N.K. and S.V.S.; writing—original draft preparation, B.Y., L.C.M., and S.P.; writing—review and editing, P.K.J., G.P.O.R., and S.V.S.; visualization, P.K.J. and B.L.T.; supervision, B.L.M. and B.S.D. All authors have read and agreed to the published version of the manuscript.

**Funding:** This research received no external funding.

**Institutional Review Board Statement:** Not applicable.

**Informed Consent Statement:** Not applicable.

**Data Availability Statement:** Not applicable.

**Acknowledgments:** The authors are grateful to the ICAR-NBSS&LUP and DST for help and support.

**Conflicts of Interest:** The authors declare that there is no conflict of interest, either financially or otherwise.

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
