# Peer review of "Mapping of Land Degradation Vulnerability in the Semi-Arid Watershed of Rajasthan, India"

_sustainability, doi:10.3390/su141610198_

Round 1

Reviewer 1 Report

I appreciate the invitation to review a research proposal that is well framed in the current lines of work. From a geographical point of view, changes in land use are one of the main scientific-methodological approaches that GIS currently have. In this sense, the present investigation correctly develops the background of knowledge on this specialty, and develops it correctly in the search for indicators of soil degradation associated with semi-arid ecosystems.

The introduction, and also the materials and methods section, are clearly developed and resolved in the research proposal.

Author Response

Dear Reviewer,

Thanks for your invaluable suggestions for improving our manuscript. The manuscript has been revised as per your suggestions. The changes are mentioned with the line number according to the revised manuscript in the response to reviewer's letter 

Sincerely,

Authors

Reviewer 2 Report

The manuscript cannot be accepted in its current state and requires major revision, especially in the validation section.

11.  The role of remote sensing and GIS in mapping land degradation is missing in the introduction.

22. There are no discussions related to the background of the multi-decision criterion method (MCDM) in the introduction section.

33. The problem statement of the current study is missing.

44. The novelty of the current research work carried out in this study is not defined in the introduction section.

55. The algorithm behind the conversion of multiple datasets of different spatial resolutions into a single spatial resolution has not been defined clearly in the manuscript.

66. The reference on which the scores were assigned in the pairwise matrix is not defined in the manuscript.

77. The normalized matrix generated in the AHP is missing in the manuscript.

88. The definitions of very low, low, moderate, high, and very high are not defined in the manuscript. There should be a clear explanation related to the definition and the range of defining the levels/classes of land degradation

     9.The validation section of the manuscript is too weak and cannot be accepted in the current state. There should be field-based direct or direct ways of validation techniques/parameters to validate the model result, merely showing the comparison with google earth images will not solve the purpose. A field survey is required at multiple random sites with indirect/direct parameters to validate the model result.

110. The conclusion section of the manuscript requires severe revision. The limitations and the future scope are missing in the current manuscript.

111.   The overall manuscript requires severe English check, as it contains many grammatical errors.

Author Response

(The authors gave the same response as above.)

Reviewer 3 Report

The authors tackle an interesting and timely topic, the mapping of land degradation vulnerability. However, the paper has many deficiencies.

Below you will find my comments in more or less descending order of importance.

Lnnn = line nnn

Title: Impressive title, but is it appropriate if we know that the actual study area is just 26 km2? I don't think so. If a reader realizes that the study area is in India, it sounds interesting, even if it is in Rajastan. But then we come down to an area of a town...

Abstract: The abstract is somewhat unbalanced: the problem statement and methodology are well proportioned, but the results are summarised in only 2 sentences in total (L30-L34). The typical reader wants to know more about the results already from the abstract.

Keywords: should be reviewed. GIS: irrelevant, AHP should be written out, remote sensing: too general

General problems:

The main deficiency (flaw?) of the paper is the handling of NDVI. As everyone knows, NDVI can change daily, but typically it has a seasonal curve, especially in semi-arid areas. As there are crops in the area (L127-L128) at least these fields MUST HAVE a seasonal variation in NDVI. The authors claim that they processed 11 years of NDVI using MODIS data with temporal resolution of 16 days (Table 1). Assuming that the area can be partially cloudy, there exist still a dozen of images yearly for which NDVI can be computed, that means ca. 100 NDVI images altogether. In Fig. 4 (a) we see "Normalized Difference Vegetation Index (2011-2022)". Either this panel title is completely wrong, or the authors did not tell us what processing resulted in this rather smooth map. (A further contradiction in L196-L197: "In the present study, MODISMOD13Q1 NDVI products were downloaded for a period of 10 years (2011-2020) using Google Earth Engine." What was then processed?) In Section 2.3.3 there is nothing about the calculation. The authors refer to a paper [25] that deals with a similar problem, but here there is no application (Theil-Sen regression, trend analysis) in the text.

I consider this as a major problem, and this raises questions about further calculations: "NDVI was selected as the second important parameter in the hierarchy" L224-L225, but we don't know what it is.

L202-L203: "Data sets were reprojected from the Sinusoidal coordinates system to the geographical coordinate system and resampled to 30 m resolution by using the bilinear interpolation technique in the QGIS." What is this? Reprojection from a METRIC system to geographical (=Latitude/Longitude) then how can you calculate 30 m grid in a geographical coordinate system. This might be an error in formulation. Please correct that. (In the original sinusoidal system you can resample it to 30 m grid.)

In my view a 100% "validation with google earth imageries" (L109) is not a proper validation. There is no ground truth data in the paper, Google Earth images can be misleading. Since the study area is very small, it could have been possible to check at least the hot spots in the field.

Structural problems:

in Introduction: several leaps of logic. The study area is not properly presented here (L70-L81), it is intercalated into the theoretical considerations.

It is just implicitly stated that "areas of the country" (L71) are in India, because in L65-L70 one can think that India is just an example for the general description of the problem.

It is explicitly stated that the study area is Bundi (Rajastan, India), in Section 2.1, if a few words are needed in the Introduction L72-L81 can placed AFTER the general introductory text. 

The last sentence of this paragraph (L81-L85) is again a general statement that has nothing to do with the study area. And then in L109-L111 the study area suddenly reappears for a single sentence (anyhow we already know that). 

Equations:

Eq(5): I am very sorry but I cannot understand this definition, especially in the light of what is written in L260-L263. Please clarify the situation/reformat the mathematical expression.

Problems with the formulation:

L92-L94: "Tools and techniques, that are more advanced such as the Analytic Hierarchy Process (AHP), Remote Sensing (RS), and Geographic Information System (GIS)..." this text is too general and too trivial (it would have been appropriate in 1990ies, but not these days).

L96-L98: "RS methods and datasets are the primary choices for modelling and assessment of land degradation, because of its accessibility, it can be assessed quickly and efficiently over a vast region" sounds like a bachelor thesis (nowadays).

Minor remarks:

L102: This is uncommon to refer to a scientist this way in a scientific paper. 

L109: "google earth" should be Google Earth

L209-L210: Please write out the units, for each cases. 

L277-L282: Please use units, here: °C.

L525: The DOI of ref [23] is erroneous: instead of a space a "/" is required.

L528: The DOI of ref [28] is erroneous: the space should be removed.

Figs. 1, 3, 4, 5: Write km instead of Kms (See webpage of BIPM, https://www.bipm.org/en/measurement-units/si-prefixes) I know that some software uses this erroneous format (shame on them!) but in a scientific paper proper SI units must be used. 

Author Response

(The authors gave the same response as above.)

Round 2

Reviewer 2 Report

Though the authors have made an attempt to validate the model using the images obtained from the field but there are not statistical analyses included in the validation of the model results. Merely using the images will not solve the purpose; statistical validation of the output will strengthen the result. There are indirect ways to statistically validate the model results. Include a statistical analysis in the validation section of the manuscript.

Author Response

Dear Editor,

Please find the response to reviewer attached,

Thanks,

Authors

Reviewer 3 Report

The authors have considerably improved the manuscript, now it has become much clearer, much more understandable, the reader can follow most of the processing details.

I still see some possibility for improvement. 

The authors inserted some new text at the end of the paper: " Thematic layers used in the study are incomparable because they all have different magnitudes. Furthermore, because each factor has a different impact on land  degradation, it can be difficult to determine the proper weight for each." (Lines 468-470) 

I do not understand this at all. It seems to me that the authors are trying to evade responsibility, which for me calls into question the whole point of the paper. I suggest that this part is seriously rewritten, because it sounds very strange in the conclusion of a scientific article. On the other hand, I don't have any problem with the rest of the text (Lines 470-474), that is a fair warning for others who wish to apply the method.

Minor remarks:

Line 173 (Eq.(1)) usually the mathematical functions are NOT written in italic (here: ln, tan is okay)

Line 357 "Weightages of sub-classes"  Weighting?

Fig 5: the sites in the map are hardly visible. Inserting arrows (just 5) can perhaps improve the visibility like in Fig 6 (which is a great improvement of the manuscript).  

Author Response

(The authors gave the same response as above.)
